# Genome-Wide Identification and Expression Analysis of the GH19 Chitinase Gene Family in Sea Island Cotton

**DOI:** 10.3390/cimb47080633

**Published:** 2025-08-07

**Authors:** Jingjing Ma, Yilei Long, Jincheng Fu, Nengshuang Shen, Le Wang, Shuaijun Wu, Jing Li, Quanjia Chen, Qianli Zu, Xiaojuan Deng

**Affiliations:** 1College of Agronomy, Xinjiang Agricultural University, Urumqi 830052, China; 17799239980@163.com (J.M.); fjc5202023@163.com (J.F.); shennengshuang@163.com (N.S.); 19882955016@163.com (L.W.); 15069495355@163.com (S.W.); 15739671550@163.com (J.L.); chqjia@126.com (Q.C.); 2College of Life Science and Technology, Xinjiang University, Urumqi 830046, China; longyilei2021@163.com

**Keywords:** *Gossypium barbadense*, chitinase, fusarium wilt, phytohormones, gene expression profile

## Abstract

In this study, GH19 chitinase (Chi) gene family was systematically identified and characterized using genomic assemblies from four cotton species: *Gossypium barbadense*, *G. hirsutum*, *G. arboreum*, and *G. raimondii*. A suite of analyses was performed, including genome-wide gene identification, physicochemical property characterization of the encoded proteins, subcellular localization prediction, phylogenetic reconstruction, chromosomal mapping, promoter cis-element analysis, and comprehensive expression profiling using transcriptomic data and qRT-PCR (including tissue-specific expression, hormone treatments, and *Fusarium oxysporum* infection assays). A total of 107 GH19 genes were identified across the four species (35 in *G. barbadense*, 37 in *G. hirsutum*, 19 in *G. arboreum*, and 16 in *G. raimondii*). The molecular weights of GH19 proteins ranged from 9.9 to 97.3 kDa, and they were predominantly predicted to localize to the extracellular space. Phylogenetic analysis revealed three well-conserved clades within this family. In tetraploid cotton, GH19 genes were unevenly distributed across 12 chromosomes, often clustering in certain regions, whereas in diploid species, they were confined to five chromosomes. Promoter analysis indicated that GH19 gene promoters contain numerous stress- and hormone-responsive motifs, including those for abscisic acid (ABA), ethylene (ET), and gibberellin (GA), as well as abundant light-responsive elements. The expression patterns of GH19 genes were largely tissue-specific; for instance, *GbChi23* was predominantly expressed in the calyx, whereas *GbChi19/21/22* were primarily expressed in the roots and stems. Overall, this study provides the first comprehensive genomic and functional characterization of the GH19 family in *G. barbadense*, laying a foundation for understanding its role in disease resistance mechanisms and aiding in the identification of candidate genes to enhance plant defense against biotic stress.

## 1. Introduction

Cotton is the main source of natural fibers for the textile industry and the mainstay of many economies worldwide; however, it is highly susceptible to aphid attacks [1]. *Gossypium hirsutum* (*G. hirsutum*), commonly known as American or upland cotton, is the most widely grown cotton species, accounting for 95% of global cotton production [2]. Only a very small fraction (less than 2%) of the global cotton output comes from other cultivated species such as *G. barbadense*, *G. arboretum*, and *G. herbaceum*. Xinjiang is the only region in China that produces island cotton, and its quality and yield are frequently affected by numerous biotic and abiotic stresses [3]. Chitin, a linear polymer of β-1,4-linked N-acetylglucosamine residues, is the second most abundant natural polysaccharide and serves as a structural scaffold in fungal cell walls, arthropod exoskeletons, and algal extracellular matrices [4,5,6]. Chitin is difficult to degrade due to its crystalline nature and because it is embedded in robust co-polymeric materials containing other polysaccharides, proteins, and minerals [7]. Plants have evolved multifaceted defense systems to counteract pathogen invasion, including cell wall reinforcement, biosynthesis of pathogenesis-related (PR) proteins, and production of antimicrobial phytochemicals and secondary metabolites [8]. Among these defense components, chitinases (Chi) and glucanases are critical PR proteins that directly target the structural components of fungal pathogens and herbivorous insects [9]. Chi are hydrolytic enzymes belonging to the glycoside hydrolase family and are found in organisms ranging from viruses to mammals. They perform diverse functions, including tissue degradation and remodeling, nutrition uptake, facilitating pathogen invasion, and regulating immune response by cleaving β-1,4-glycosidic bonds in chitin in response to abiotic and biotic stresses [10]. The GH19 family chitinases, initially identified as unique defense proteins in higher plants, play a central role in the plant’s antifungal immunity [11]. The systematic classification of plant chitinases has undergone a gradual evolution from the early classification based on domain differences (Classes I-III), to genome-driven sequence homology expansion (Classes IV-V), and more recently to functional genomics-based subdivision (Classes VI-VII) (Figure 1). For example, Streptomyces corchorusii strain UCR3-16, isolated from rice rhizospheric soils, exhibits antifungal activity against rhizospheric soils, exhibits antifungal activity against 6 major rice fungal pathogens by producing diffusible and volatile compounds, including enzymes that degrade fungal cell walls [12].Substantial evidence indicates that plants have evolved two main innate immune mechanisms in response to infection by various pathogens [13,14]. The first layer of inducible immunity is mediated by surface-localized pattern recognition receptors that bind microbe- and damage-associated molecular patterns, thereby activating pattern-triggered immunity [15]. Conversely, most intracellular immune receptors are nucleotide-binding leucine-rich repeat receptor proteins that specifically recognize pathogen-secreted effectors to mediate effector-triggered immunity [16]. This immune response is often accompanied by rapid and localized programmed cell death, known as hypersensitive response. After localized invasion by bacterial pathogens, plants develop systemic acquired resistance in uninfected tissues. This inducible defense can be triggered by local infection or treatment with elicitors [17], resulting in enhanced resistance to a broad range of subsequent pathogens. For instance, transgenic overexpression of the CHITINASE 2 gene (*LcCHI2*) from *Leymus chinensis*, which encodes a class II Chi, in tobacco confers enhanced salt tolerance, as evidenced by reduced sodium accumulation, lower malondialdehyde content, and decreased electrolyte leakage under saline conditions [18]. Similarly, osmotic stress and abscisic acid (ABA) signaling have been shown to promote the expression of the *ptch28* Chi gene in transgenic tobacco and maize [19]. Although plants do not contain chitin, they can produce Chi to protect themselves from biotic and abiotic stressors through a combination of constitutive and inducible defense mechanisms. Notably, these dual recognition mechanisms operate synergistically to form an intricate and sophisticated immune surveillance network in plants.

In addition to their well-established role in plant defense against pathogens, Chi has also been implicated in abiotic stress response, helping plants survive in harsh environments [20]. For instance, a cold-responsive Chi gene, *BiCHT1*, was isolated from bromegrass (*Bromus inermis Leyss*.) ‘Manchar’ suspension cells, and while *BiCHT1* is expressed at low levels under normal conditions, its transcript accumulates significantly after exposure to low temperatures (10 °C and 4 °C) [21]. *Chit42* and *Chit33* from *T. harzianum* CECT 2413, which lack a chitin-binding domain, play an important role in this strain’s biocontrol activity against plant pathogens owing to their high hydrolytic activity on insoluble substrates, such as chitin-rich fungal cell walls [22]. In transgenic tobacco (*Nicotiana tabacum* L.), overexpression of either *CHIT33* or *CHIT42* has been shown to confer broad resistance to fungal and bacterial pathogens, as well as tolerance to salinity and heavy metals, with no obvious detrimental effects on plant growth [23]. Additionally, treatment of lichen thalli with hydrolytic enzymes, particularly Chi and lichenase, enhanced melanin production. The proposed mechanism is that melanin binds firmly to hyphal cell wall carbohydrates (especially chitin and 1,4-β-glucans), thereby strengthening the melanized upper cortex of lichen thalli and improving survival under UV stress [24]. Plants rapidly reprogram Chi gene expression through phytohormone signaling networks to combat environmental stresses [25,26]. Biotic (e.g., pathogen infection) and abiotic challenges trigger the biosynthesis of key phytohormones, including jasmonic acid (JA), ethylene (ET), ABA, salicylic acid (SA), and gibberellin (GA), which act as master regulators of Chi-mediated defense responses [27,28]. Consequently, plants rapidly reprogram Chi gene expression through these phytohormone signaling networks to combat environmental stresses. Furthermore, SA signaling, UV-B irradiation, and mechanical wounding have been shown to induce the accumulation of transcripts and corresponding proteins of a class-III Chi gene (IF3) in *Lupinus luteus*, thereby enhancing the plant’s resistance to biotic and abiotic stress [29,30]. In *Arabidopsis thaliana*, the expression level of *AtChiC* is significantly upregulated by ABA and JA, and the strawberry Chi gene *FnCHIT2* is highly induced by SA; notably, the ectopic expression of *FnCHIT2* in Arabidopsis enhances resistance to fungal pathogens [31,32].

The recent availability of genomic sequences for multiple cotton species (including *G. barbadense*, *G. hirsutum*, *G. arboreum*, and *G. raimondii*) provides an opportunity for comparative analyses of important gene families. For example, chitinases belong to two glycosyl hydrolase families: GH18 and GH19 [33]. GH19 chitinases are widespread in bacteria, microsporidia, and plants, where they function in nutrient acquisition or defense, particularly during plant–pathogen interactions [34]. Interestingly, a novel multifunctional GH19 chitinase (*CaChi19A*) containing three chitin-binding domains was recently cloned from *Chitinilyticum aquatile* CSC-1; this enzyme retains high activity at low temperatures and shows promise for the biocontrol of fungal diseases in agriculture [35].

However, the functional evolution of Chi family genes in relation to *Fusarium wilt* resistance in *G. barbadense* remains largely unexplored. Based on the established role of GH19 Chi in fungal defense and hormone signaling [36], we hypothesized that certain GbChi isoforms in the wilt-resistant cotton cultivar 06-146 are specifically specialized for resistance to *Fusarium oxysporum* via hormone-mediated signaling pathways. To address this hypothesis, we identified the Chi gene family across four cotton species (*G. barbadense*, *G. hirsutum*, *G. arboreum*, and *G. raimondii*) and conducted a comprehensive comparative analysis. Specifically, we classified genes using phylogenetic methods and characterized their physicochemical properties, chromosomal locations, exon-intron organization, and predicted subcellular localizations. We also constructed co-expression networks to examine the regulatory dynamics of GbChi genes in response to various hormone treatments and pathogen challenges. Our primary objectives were as follows: (1) systematically characterize the structural and evolutionary features of GH19 Chi in *G. barbadense*; (2) identify *Fusarium*-responsive GbChi candidate genes with putative roles in wilt resistance; and (3) validate the functional link between hormone signaling and pathogen-induced GbChi expression. Collectively, the genomic and functional insights gained from this study, derived from extensive analyses leveraging open-source tools, provide a foundation for elucidating cotton disease resistance mechanisms and guiding future strategies for gene use.

## 2. Materials and Methods

### 2.1. Identification and Physicochemical Characterization of Gene Family Members

Genome assemblies of *Gossypium hirsutum* (TM-1, ZJU-AD1_v2.1_a1.0), *G. barbadense* (Hai7124, HAU_v2.0), *G. arboreum* (CRI-A2_v1.0), and *G. raimondii* (JGI_221_v2.1) were retrieved from the CottonGen database (http://www.cottongen.org, accessed on 1 December 2024), which provides access to publicly available genomic, genetic, and breeding data for cotton. In parallel, the Pfam database (https://pfam.xfam.org/, accessed on 1 December 2024) was queried to obtain the hidden Markov model (HMM) profile for the GH19 Chi family (Pfam ID: PF00182). This HMM profile was used to perform a local search with HMMER v3.0 against all protein sequences using an E-value threshold of 0.001. All significant hits were considered candidate Chi genes, and redundant sequences were removed through manual curation.

### 2.2. Physicochemical Profiling and Subcellular Localization Prediction

Physicochemical properties of each Chi protein (including amino acid sequence length, theoretical isoelectric point, molecular weight (Mw), instability index, aliphatic index, and grand average of hydropathicity) were calculated using the ProtParam tool (https://web.expasy.org/protparam/, accessed on 7 December 2024).

Subcellular localization of each Chi protein was predicted using the Cell-PLoc 2.0 platform (http://www.csbio.sjtu.edu.cn/bioinf/Cell-PLoc-2/, accessed on 12 December 2024). This platform integrates multiple machine learning algorithms with species-specific training datasets to infer protein compartmentalization.

### 2.3. Phylogenetic Reconstruction and Chromosomal Mapping

Chi protein sequences from the four Gossypium species (*G. hirsutum*, *G. barbadense*, *G. arboreum*, and *G. raimondii*) were subjected to phylogenetic analysis. Multiple sequence alignment was performed using Clustal Omega (v1.2.4), and a Neighbor-Joining (NJ) tree was constructed in MEGA11 (v11.0.13) with 1000 bootstrap replicates to assess node support. The resulting tree topology was visualized and annotated using the Interactive Tree of Life (iTOL) platform (https://itol.embl.de/, accessed on 16 December 2024).

The chromosomal locations of Chi genes were determined by parsing genome annotation files to extract their coordinates. Chromosomal density profiles and gene distribution maps were then generated using TBtools (v1.108) with default parameters.

### 2.4. Structural Characterization and Cis-Regulatory Element Profiling

The promoter regions (~2000 bp upstream of the ATG start codon) of the *G. barbadense* Chi genes were extracted and analyzed using TBtools (v1.108). Regulatory elements within these regions were identified using PlantCARE (https://bioinformatics.psb.ugent.be/webtools/plantcare/html/, accessed on 23 December 2024) and then categorized into stress-responsive, phytohormone-related, developmental, and light-responsive motifs. The resulting data were visualized using the Basic Biosequence View and HeatMap modules of TBtools (v1.108), facilitating a comparative analysis of regulatory element enrichment across the GbChi gene family.

### 2.5. Synteny and Collinearity Analysis

Intragenomic synteny within *G. barbadense* and inter-genomic collinearity across Gossypium species (*G. hirsutum*, *G. arboreum*, and *G. raimondii*) were analyzed using Multiple Collinearity Scan toolkit (MCScanX) with the default parameters. Homologous genomic blocks and duplicated gene pairs were identified through pairwise genome alignments and visualized using TBtools (v1.108) to highlight the evolutionary conservation and lineage-specific duplication events.

### 2.6. Tissue-Specific Expression Profiling of GbChi Genes

Fragments per kilobase of transcription per million mapped reads (FPKM) values for *G. barbadense* Chi genes across diverse tissues were retrieved from the Cotton Functional Genomics Database (CottonFGD, https://cottonfgd.org/, accessed on 26 December 2024). The resulting expression matrix was normalized and hierarchically clustered using the HeatMap module of TBtools (v1.108), revealing spatiotemporal expression patterns and tissue-specific regulatory differences associated with plant development and growth.

### 2.7. Expression Profiling of GbChi Genes Under Hormonal Treatments and Biotic Stress

Seeds of *G. barbadense* cv. 06-146 (a wilt-resistant cultivar [37]) were surface-sterilized with 0.5% sodium hypochlorite (NaClO) and thoroughly rinsed with sterile distilled water. The sterilized seeds were placed in Petri dishes on moistened filter paper and allowed to germinate for 2 d until the radicles emerged. The germinated seedlings were then transplanted into sterile soil and grown under a 16 h light/8 h dark photoperiod at 25 °C for 21 d. At the three-leaf stage, the seedlings were subjected to various treatments to assess GbChi gene expression under hormonal and biotic stress. For hormonal treatments, the leaves were sprayed with solutions of ethylene (applied as ethephon, 100 μM/L), gibberellic acid (GA_3_, 50 μM/L), and ABA (100 μM/L). For biotic stress, roots were inoculated with *Fusarium oxysporum* f. sp. *vasinfectum* (race 7) at approximately 1 × 10^7^ spores/mL. Each treatment group consisted of three biological replicates, each containing tissues pooled from 5 individual plants. Tissue samples were collected over a 72 h course following treatment initiation. For the pathogen-inoculated plants, composite root, stem, and leaf samples were harvested at 0, 12, 24, 36, 48, and 72 h after inoculation. Leaf samples were collected from hormone-treated plants at the same six time points. Total RNA was isolated from each sample using a polyphenol-polysaccharide optimized kit (Tiangen Biochemical Technology (Beijing) Co., Ltd., Beijing, China), and first-strand cDNA was synthesized using the PrimeScript™ RT Reagent Kit (ABM Biotech, Zhenjiang, China; parent company: Applied Biological Materials Inc., Canada). Quantitative real-time PCR (qRT-PCR) was performed on a QuantStudio 5 platform using SYBR Green chemistry with GbUBQ7 as the internal reference gene [38] Relative expression levels were calculated using the 2-ΔΔCt method. All qRT-PCR reactions were performed in triplicate for each sample. This integrative experimental design enabled the correlation of specific hormone treatments and pathogen infection with GbChi gene expression patterns, thereby elucidating the dynamics of Chi induction.

## 3. Results

### 3.1. Physicochemical Characterization of Chi Family Members in Four Cultivated Cotton Species

A genome-wide analysis identified 107 Chi genes across four cultivated cotton species (Appendix A), including 35 in *G. barbadense*, 37 in *G. hirsutum*, 16 in *G. raimondii*, and 19 in *G. arboreum*. These genes were unevenly distributed across the chromosomes, with preferential localization on homologous chromosomes A01, A03, A06, A09, A10, and A13, and their corresponding D-subgenome counterparts.

Analysis of the protein sequence characteristics demonstrated substantial diversity within the gene family. The encoded Chi proteins ranged from approximately 100 to 600 amino acids in length, and their predicted molecular masses varied widely from 30 to 32 kDa. For example, the smallest member (*GhChi22*) had a predicted molecular mass of ~9.9 kDa, whereas the largest (*GhChi38*) was ~97.3 kDa. Furthermore, subcellular localization predictions indicated that most of these proteins (82.6%) were likely extracellular. Smaller fractions were predicted to localize to the lysosome (9.1%), nucleus (5.4%), and plasma membrane (2.9%).

### 3.2. Phylogenetic Reconstruction and Chromosomal Localization

Chi family members were systematically annotated in each species: *GbChi1–GbChi35* in *Gossypium barbadense*, *GhChi1–GhChi37* in *G. hirsutum*, *GaChi1–GaChi19* in *G. arboreum*, and *GrChi1–GrChi16* in *G. raimondii*. Phylogenetic analysis of all 107 Chi proteins revealed three primary clades across the four cotton species (Figure 2). This finding indicates that clade composition is largely conserved among these species, despite differences in the number of Chi genes within each lineage. This phylogenetic topology suggests a conserved pattern of functional diversification during the evolution of Gossypium. In particular, clade-specific expansions or contractions of the Chi gene family likely reflect lineage-specific adaptations.

Additionally, conserved domain analysis (Figure 3) identified three signature motifs present in most Chi proteins: CHIT_BIND_I_1 (a chitin-binding domain), Chi family 19 signature 2 (the catalytic core), and CHITINASE_19_1 (a substrate specificity determinant). However, these canonical motifs were not detected in *GbChi11*, *GbChi15*, and *GbChi32*, possibly due to the evolutionary divergence in the motif architecture of these genes.

Species-specific chromosomal distribution patterns were also observed (Figure 4). In *G. barbadense* and *G. hirsutum*, Chi genes were distributed across 12 chromosomes, whereas in *G. arboreum* and *G. raimondii*, they were found on only five chromosomes. Across all four species, most Chi genes were located in conserved syntenic clusters, with only a few singletons. Notably, single-copy Chi genes were observed on chromosome 06 (Chr06) in both tetraploid (A-and D-subgenomes) and diploid cotton genome. This conserved chromosomal architecture implies that these loci were retained during cotton polyploidization.

### 3.3. Characterization of Chi Gene Family

In the *G. barbadense* genome (which contains 35 GbChi genes), intragenomic synteny analysis identified 25 collinear (duplicated) gene pairs (Figure 5). These duplicated pairs showed distinct distribution patterns between the two subgenomes. For example, two duplicated pairs (*GbChi10/GbChi12* and *GbChi7/GbChi15*) were located on the A subgenome (chromosomes A09 and A10), whereas four pairs (*GbChi17/GbChi18*, *GbChi19/GbChi21*, *GbChi35*, and *GbChi28/GbChi31*) were located on the D subgenome (chromosomes D01, D02, and D09–D11).

Moreover, cross-species collinearity analysis demonstrated pronounced conservation between *G. barbadense* and *G. hirsutum*, with these two tetraploid species sharing 68 syntenic gene pairs (Figure 6). In contrast, moderate conservation was observed between *G. barbadense* and its diploid progenitors. Specifically, *G. barbadense* shared 34 syntenic pairs with *G. arboreum* (A-genome ancestor) and 32 with *G. raimondii* (D-genome ancestor). Notably, several chromosomal hotspots in *G. barbadense* exhibited one-to-many syntenic relationships. For instance, four GbChi loci (*GbChi3*, *GbChi10*, *GbChi21*, and *GbChi30*), all located on chromosomes Chr09–11, exhibited pleiotropic collinearity across multiple chromosomes in other cotton genomes. Interestingly, a unique syntenic link was identified between *GbChi20* (in *G. barbadense*) and *GaChi3* (in *G. arboreum*), suggesting the lineage-specific retention of this gene pair.

### 3.4. Cis-Acting Elements in Promoter Region of GbChi Gene Family

Promoter analysis of GbChi genes in *G. barbadense* revealed a diverse array of cis-regulatory elements within their 2000 bp upstream regions, with stress and hormone-responsive motifs being dominant (Figure 7). Antioxidant response elements (ARE) were the most prevalent, present in 92.4% of these promoters, and light-responsive Box4 motifs were detected in 87.5%. ABA-responsive elements were also common (found in 78.3% of promoters), highlighting the evolutionary emphasis on environmental stress adaptation. In contrast, promoter elements associated with development (such as the CAT-box and HD-Zip1 motifs) were sparse, found in only 16 of the GbChi promoters. Meanwhile, MYB-binding sites related to flavonoid biosynthesis and drought response were present in several promoters, underscoring the potential dual roles of some GbChi genes in both biotic and abiotic stress responses. This cis-regulatory landscape, rich in stress-responsive elements but relatively poor in growth-related motifs, reflects the functional specialization of the GbChi genes. This suggests that these genes are more geared toward enhancing stress resilience than regulating growth, which is a hallmark of cotton adaptation to stress-prone ecosystems.

### 3.5. Spatiotemporal Expression Profiling of GbChi Genes in Gossypium barbadense

To elucidate the organ-specific regulatory roles of the Chi gene family in *G. barbadense*, we analyzed transcriptomic data from seven different tissues (root, stem, leaf, bract, flower, receptacle, and sepal) using hierarchical clustering (Figure 8). This analysis grouped the GbChi genes into four distinct clades, each of which appeared to have extensive functional specialization. Clade 1 contained *GbChi23*, which showed pronounced transcriptional activity and was predominantly expressed in bracts (with notable expression in roots), suggesting a potential role in the defense of floral organs. Clade II was characterized by expression mainly in leaves and bracts, with *GbChi5* and *GbChi14* particularly enriched in vascular bundles and a secondary activation observed in sepals. These patterns imply that clade II genes play a dual role in pathogen interception and in structural reinforcement. Clade III genes (*GbChi19*, *GbChi21*, and *GbChi22*) exhibited consistently high expression in both roots and stems, suggesting a constitutive role in root-shoot communication and the maintenance of lignocellulosic integrity. Finally, clade IV, represented by *GbChi29*, was enriched in floral organs but paradoxically exhibited repressed expression levels. This pattern may reflect a developmental trade-off between reproductive defense and resource allocation. Overall, this clade-specific expression dichotomy underscores the functional diversification of the GbChi gene family: clades I and II coordinate organ-preferential defense responses, clade III maintains vegetative structural homeostasis, and clade IV mediates context-dependent immune responses in the reproductive tissues.

### 3.6. Hormonal Regulation of GbChi Gene Expression

The promoter regions of GbChi genes contain multiple cis-regulatory elements associated with phytohormone responsiveness (Figure 9), suggesting possible crosstalk between hormonal signaling and the transcriptional regulation of these genes. To investigate this interplay, we selected 12 GbChi genes for expression analysis following various hormone treatments.

Hormonal signaling often intersects with plant immune mechanisms. For example, nucleotide-binding leucine-rich repeat (NLR) immune receptors trigger a hypersensitive response against viruses, such as potato virus [39], tobacco necrosis virus [40], and Chinese wheat mosaic virus [41], which can be modulated by hormones. In this context, we first examined the effect of ABA, a master regulator of stress adaptation, on the expression of GbChi. Our qRT-PCR profiling revealed that nine GbChi genes (*GbChi1/5/7/9/12/15/17/24/30*) were progressively upregulated under ABA treatment, with expression peaking at 48 h post-induction (5.8–14.3-fold increase). In contrast, *GbChi4*, *GbChi6*, and *GbChi29* showed only transient induction, peaking at 12 h (~3.2–6.7-fold above baseline) before declining. This transient response suggests possible feedback inhibition or desensitization of ABA signaling receptor activity.

Next, we assessed GbChi gene responses to ethylene, a pleiotropic phytohormone that regulates growth, senescence, and stress adaptation, to determine whether its regulation involves ethylene-insensitive3 (EIN3)/ethylene response factor (ERF)-mediated transcriptional cascades (Figure 10). Upon treating *G. barbadense* seedlings with 100 μM ethephon (an ethylene-releasing compound), two distinct expression patterns were observed for the GbChi genes. Seven genes (*GbChi1/15/17/18/25/33/35*) were progressively induced, with a 4.2–9.8 fold increase in expression by 48 h post-induction. This sustained induction suggests the presence of ethylene-responsive GCC-box motifs in their promoters, which is a hallmark of canonical EIN3-mediated activation. Conversely, *GbChi6* and *GbChi23* were rapidly repressed after 8 h, with expression dropping to ~0.3–0.5 fold of the baseline. The presence of ERF4 repressor-binding sites in their promoters suggests the competitive suppression of ethylene signaling in these genes. This dichotomy likely reflects adaptive partitioning, with ethylene-inducible GbChi members primarily belonging to defense-specialized clades I/II, whereas the suppressed members belong to growth-associated clade III. In other words, ethylene signaling appears to promote defense-related GbChi isoforms while inhibiting those associated with growth processes, highlighting the trade-off in hormonal crosstalk.

Finally, to examine whether GA signaling through the DELLA–GID1 pathway regulates GbChi gene expression, we treated the seedlings with 50 μM GA_3_. GA_3_ treatment revealed three distinct expression patterns among the treated GbChi genes (Figure 11). Sustained inducers, *GbChi4/12/17/18/35*, were progressively upregulated, peaking at 72 h post-induction (3.5–8.2-fold increases). This prolonged induction correlated with the presence of GARE motif sequences (ACGTGKC; *p* < 0.01) in their promoters, which recruit DELLA-TF complexes for sustained activation. Biphasic suppressors, *GbChi6/23/25/29* exhibited a biphasic response: an initial transient induction (1.8–2.4-fold at 4 h), followed by a drop below baseline expression (0.4–0.7-fold by 72 h). This pattern reflects GA’s dual effect, with early priming of gene expression followed by feedback attenuation, likely through DELLA–SLY1-mediated ubiquitination and proteasomal degradation of signaling components. The oscillatory regulators, *GbChi1* and *GbChi33*, displayed oscillatory expression patterns. *GbChi1* showed three distinct peaks at 8, 24, and 48 h post-induction, whereas *GbChi33* peaked at approximately 4.7-fold at 24 h before declining. These pronounced oscillations suggest GA-responsive chromatin dynamics (e.g., looping or nucleosome repositioning), which produce pulsatile transcriptional outputs. This tripartite regulatory scheme illustrates GA’s capacity to fine-tune Chi gene activity across different time scales, balancing growth-promoting and defense-optimizing outcomes. Notably, the divergent kinetic profiles under GA treatment correspond to the phylogenetic groupings of the genes: the sustained inducer genes reside predominantly in stress-adapted clades I/II, whereas the suppressor-type genes cluster in the developmental clade IV. This alignment mirrors GA’s dual role in plant defense (e.g., against Fusarium pathogens) and growth (internode elongation), suggesting that different GbChi isoforms help balance growth-defense trade-offs through their specific sensitivities to GA.

### 3.7. Transcriptional Reprogramming of GbChi Genes in Response to Fusarium Wilt

To examine Chi-mediated defense mechanisms against biotic stress, we profiled the transcriptional dynamics of GbChi genes in *G. barbadense* following inoculation with *Fusarium oxysporum* f. sp. *vasinfectum*. Our temporal transcriptional analysis revealed three distinct response modalities (Figure 12). (1) Progressive activators (*GbChi1*, *GbChi4*, *GbChi17*, and *GbChi18*) displayed sustained upregulation, reaching a 4.8–6.3-fold increase by 72 h post-inoculation (hpi). This induction was correlated with fungal biomass accumulation (R^2^ = 0.87–0.93, *p* < 0.001). The enrichment of MYC2 and MYB44 binding sites in the promoters of these genes suggests that defense potentiation is mediated by JA and ethylene signaling. In contrast, early responders, such as *GbChi15*, *GbChi23*, *GbChi29*, and *GbChi33*, showed a rapid induction (3.5–7.2-fold within 2–4 hpi) followed by a gradual decline, indicating PR protein-mediated recognition of pathogen-associated molecular patterns (PAMP). *GbChi6*, categorized as a constitutive suppressor, exhibited persistent downregulation (0.2–0.4-fold of the baseline). This downregulation may result from pathogen-mediated manipulation of host RNA silencing pathways via fungal sRNA-induced *GbChi6* mRNA cleavage.

Mechanically, this bifurcated dynamic response appears to reflect the evolutionary optimization of defense responses. Early responders initiate chitin oligomer perception via lysin motif (LysM) receptor kinases, whereas progressive activators reinforce the cell walls through callose deposition. The paradoxical downregulation of *GbChi6* may represent a fungal counter-defense strategy aimed at evading the host’s β-1,3-glucanase activity. This finding underscores the co-evolutionary arms race in cotton–Fusarium interactions. Consistent with these results, weighted gene co-expression network analysis (WGCNA) placed *GbChi1/4/17/18* in a “Defense-Lignification” module, whereas *GbChi6* was grouped in a “Growth-Primacy” module (ME4). This clustering further corroborates the functional divergence among the GbChi gene responses.

## 4. Discussion

Chi are enzymes that hydrolyze chitin and can be phylogenetically classified into seven subgroups (GH18, GH19, GH20, GH25, GH48, GH75, and GH85) based on their amino acid sequence similarities. The GH19 subgroup is a pivotal clade within the Chi superfamily. The Chi gene family has been identified and analyzed in diverse plant species [42,43,44,45,46,47,48,49]. In the present study, time-calibrated phylogenetic analysis revealed three primary clades. The GH19 clade comprised 107 conserved members across the tetraploid and diploid cotton species examined, including *GbChi1–GbChi35* in *G. barbadense*, *GhChi1–GhChi37* in *G. hirsutum*, *GaChi1–GaChi19* in *G. arboreum*, and *GrChi1–GrChi16* in *G. raimondii*. Notably, GH19-encoded proteins are relatively low in molecular weight (30–32 kDa) and structurally simple. These features may have conferred evolutionary advantages in primordial biological environments by facilitating spontaneous chemical interactions and biosynthetic processes. Taken together, these findings suggest that the Chi gene family has remained relatively conserved throughout cotton genome evolution.

Gene duplication is a prevalent phenomenon and a major driving force in plant genome evolution. It occurs in multiple forms, including tandem [50], segmental [51], and whole-genome duplications [52]. For example, a comparative genomic study of *Cyclocarya paliurus* revealed two rounds of recent whole-genome duplication and identified 691 genes with dosage effects that likely contributed to its adaptive evolution. In our analysis of 35 GbChi genes from *G. barbadense*, gene duplication was also common: we identified 25 collinear gene pairs, including two pairs in the A subgenome (on chromosomes A09 and A10) and four pairs in the D subgenome (on chromosomes D01, D02, D09, and D11). Although most Chi genes were found in conserved syntenic clusters, we observed singleton genes on Chr06 in both tetraploid (A and D subgenomes) and diploid cotton lineages. This pattern implies the ancestral retention of these functional loci during polyploidization in Sea Island cotton. Notably, the molecular weight and isoelectric point of a protein are important factors that influence its subcellular localization and function [53]. Our analysis revealed substantial variation in both the molecular weight and isoelectric point among the GbChi proteins. Consistently, subcellular localization predictions indicated that these enzymes are predominantly extracellular. This spatial distribution aligns with prior observations of Chi localized to the cell wall, such as *Pnchi1* from *Panax notoginseng* and *FvChi-14* from wood strawberry (*Fragaria vesca*). Both enzymes have been shown to enhance pathogen resistance when overexpressed in *Arabidopsis thaliana* [54]. These findings suggest potential environment-dependent functional diversification within this protein family.

Phylogenetic reconstruction of the GH19 genes in the four Gossypium species revealed that they clustered closely together in the dendrogram (Figure 2), indicating their significant evolutionary conservation. This topological congruence supports a shared ancestral origin for GH19 genes and highlights their functional preservation throughout the evolutionary history of cotton. In addition, chromosomal mapping demonstrated that GH19 genes were unevenly distributed across the genomes of the four cultivated cotton species. This genomic asymmetry correlates with previously reported lineage-specific evolutionary dynamics [55] and provides important insights into the GH19 family evolutionary history in *Gossypium*. Concurrently, a conserved domain analysis of GH19 proteins in *G. barbadense* confirmed three invariant domains: CHIT_BIND_I_1 (chitin-binding motif), Chi family 19 signature 2 (catalytic core), and Chi_19_1 (substrate interaction interface). Motif analysis revealed that *GbChi11*, *GbChi15*, and *GbChi32* lacked specific canonical structural features. Typically, evolutionarily conserved domains are maintained through strong purifying selection. The loss of these canonical motifs in specific GbChi members represents this structural divergence. This divergence may have arisen from evolutionary drift within the GH19 family and may have contributed to functional diversification. Collectively, these findings support the hypothesis that extracellularly localized Chi acts as a frontline defense component. They may do so by recognizing and binding PAMPs on the surface of invading microorganisms, thereby initiating defense responses known as pattern-triggered immunity.

Chi expression and enzymatic activity exhibit species and organ-specific variations across plants. For example, in potato (*Solanum tuberosum*), Chi activity and transcription are markedly elevated in senescent leaves and roots, with vascular tissues and stomatal complexes in mature leaves, young internodes, and petioles showing particularly high accumulation [56]. In contrast, tobacco (*Nicotiana tabacum*) shows higher Chi levels in the basal leaves and roots, with minimal content in the apical leaves. In *Arabidopsis thaliana*, transcripts of *AtCTL2* are predominantly localized to stems, whereas *AtCTL1* is broadly expressed across organs, implying distinct isoform localization [57]. Similarly, in sugarcane (*Saccharum officinarum*), *ScChi* is preferentially expressed in the leaf and stem epidermal layers, whereas *ScChiVIII* transcripts are significantly enriched in buds compared to those in roots, stems, and leaves [58]. Transcriptomic analysis revealed that GbChi genes in *G. barbadense* are predominantly expressed in the roots and stems, with elevated transcript levels in sepals and complex regulatory patterns in floral tissues. In contrast, *GbChi24* was highly expressed in floral organs. The overall low expression of certain GbChi paralogs in reproductive tissues implies that these genes prioritize vegetative organ functions over floral development. These tissue-specific expression profiles underscore the regulatory complexity of Chi genes during plant growth, reflecting functional divergence across organs and their coordinated contributions to *G. barbadense* developmental processes.

Many stress-related regulatory elements are abundantly enriched in the promoters of GH19 family Chi genes. Similarly, cis-acting regulatory elements serve as important molecular switches in transcriptional networks that control various biological processes, including abiotic stress responses, hormone signaling, and developmental programs [59]. Numerous studies have demonstrated that Chi genes play significant roles in stress responses [60,61,62]. Consistent with these findings, our promoter analysis showed that the promoter regions of GbChi genes harbored multiple cis-regulatory elements associated with phytohormone responsiveness, including antioxidant-response elements (present in 92.4% of these promoters), light-regulated Box4 motifs (87.5%), and ABA-responsive elements (78.3%) in *G. barbadense*. Our hormone induction experiments revealed differential expression patterns among the GbChi genes under ABA, ET, and GA treatment. For instance, *GbChi4/6/29* showed a transient increase in transcript levels in response to ABA, peaking at 12 h and gradually declining thereafter. This transient induction is consistent with the rapid ABA-mediated defense response observed in plant antiviral immunity [63,64]. In contrast, *GbChi1/15/17/18* and several other GbChi genes were continuously upregulated under prolonged ethylene treatment, reflecting the role of ethylene in sustained defense, which is consistent with its known long-term regulatory functions in leaf senescence and systemic resistance (SAR) [65]. In addition, ethylene may stabilize Chi gene expression through EIN3/EIL1 transcription factors [66], thereby enhancing the cell wall repair ability against *Botrytis cinerea* infection. Notably, *GbChi6* was downregulated by both ABA and ethylene, suggesting that this gene might be involved in a growth–defense trade-off or participate in antagonistic crosstalk with other hormone signaling pathways. Under GA treatment, most GbChi genes (e.g., *GbChi4/12/17/18*) were upregulated, whereas *GbChi6/23* were downregulated. In *Arabidopsis*, the absence of GA suppresses GA-responsive genes due to the accumulation of DELL proteins (a GRAS-family repressor) via JA/ethylene. Thus, the GA-induced downregulation of *GbChi6* in cotton may reflect a temporary dampening of certain defense genes as a result of DELLA protein dynamics [67]. Accordingly, the effect of GA on GbChi expression is likely related to GA-mediated DELLA protein degradation. We observed that the transcripts of *GbChi4/12/17/18* and others peaked at 72 h under GA treatment, suggesting that these genes may contribute to long-term defense responses, such as callose synthesis and targeted lignin deposition in the cell walls. Overall, members of the GbChi family exhibited distinct transcriptional responses to ABA, ET, and GA, indicating that each hormone orchestrates specific aspects of plant defense responses. On the other hand, different plant Chi genes have been linked to varying levels of *Fusarium wilt* resistance [68,69]. In our study, we observed three distinct transcriptional response modes to *Fusarium oxysporum* (the causative agent of *Fusarium wilt*) at 72 h post-inoculation: progressive activators, sustained upregulation (e.g., *GbChi1*, *GbChi4*, *GbChi17*, and *GbChi18*); early responders, rapid early induction (e.g., *GbChi15*, *GbChi23*, *GbChi29*, and *GbChi33*); and constitutive suppressors, persistent downregulation (e.g., GbChi8). These response patterns highlight the co-evolutionary arms race dynamics in cotton–*Fusarium* interactions. Together, our comprehensive genomic and functional analyses of the GH19 Chi family, coupled with the expression profiling of GbChi genes under various phytohormone treatments and *Fusarium* wilt infection, lays a solid foundation for future studies on Chi genes in cotton.

## 5. Conclusions

This study systematically identified the GH19 Chi gene family in Sea Island cotton (*Gossypium barbadense*) and revealed its key roles in disease resistance and hormone response pathways in cotton. Phylogenetic analysis showed that GH19 Chi segregated into three primary clades with a consistent grouping pattern across different Gossypium species. Subcellular localization predictions indicated that most GH19 proteins are localized in the extracellular space, with only a few members targeted to the lysosomes, nuclei, or plasma membranes. Genomic mapping revealed that GH19 genes are distributed across 12 chromosomes in the tetraploid cottons *G. barbadense* and *G. hirsutum* (Sea Island and upland cotton, respectively), whereas they reside on only five chromosomes in the diploid species *G. arboreum* and *G. raimondii*. Promoter analysis showed that the promoters of GH19 genes in *G. barbadense* are enriched with light-responsive elements and contain numerous hormone- and abiotic stress–responsive cis-elements. Moreover, GH19 genes displayed distinct organ-specific expression patterns in *G. barbadense*. Each gene responded to different hormone treatments and *Fusarium wilt* infection to varying degrees, underscoring the importance of GbChi genes in cotton defense against biotic stress. Although the current study was limited to genomic and transcriptomic analyses, we identified several GH19 genes, such as GbChi17, GbChi18, and GbChi23, that were highly responsive to hormonal signals and pathogen challenges. These genes have been designated as priority targets for future functional studies to validate their roles and potentially exploit them to improve disease resistance in cotton plants.

## Figures and Tables

**Figure 1 cimb-47-00633-f001:**
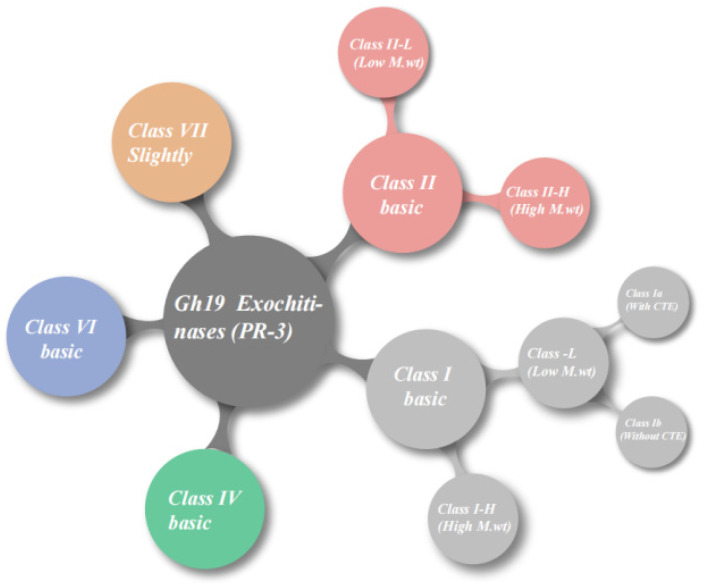
Comprehensive classification of plant chitinases.

**Figure 2 cimb-47-00633-f002:**
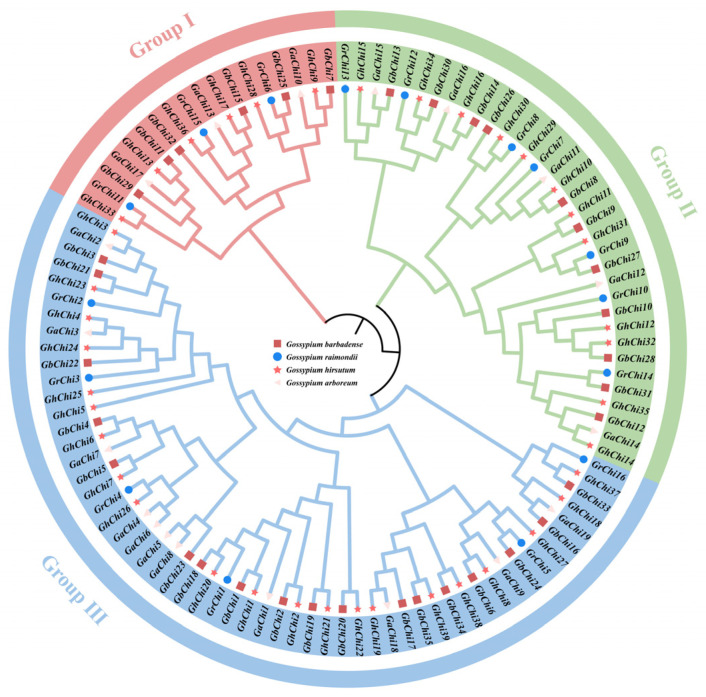
Phylogenetic tree of 107 Chi genes in four cotton cultivars.

**Figure 3 cimb-47-00633-f003:**
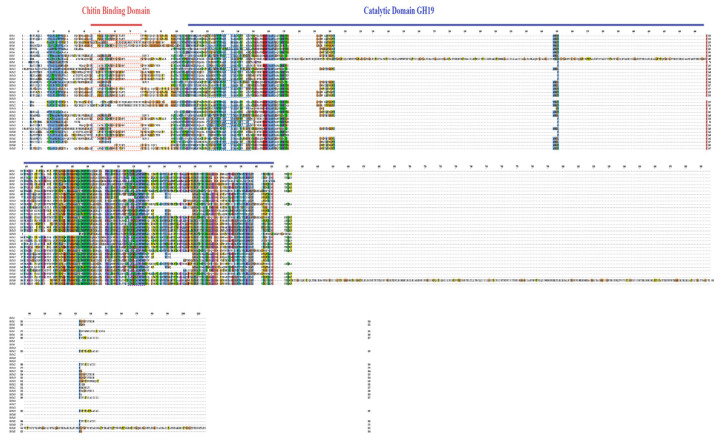
Comparison of multiple sequences of conserved domains of members of the island cotton GbChi gene family. Notes: Red-dashed box indicates feature of CHIT_BIND_I_1 (PS00026, C-x(4,5)-C-C-S-x(2)-G-x-C-G-x(3,4)-[FYW]-C); blue-dashed box shows chitinase family 19 signature 1 (PS00773, C-x(4,5)-F-Y-[ST]-x(3)-[FY]-[LIVMF]-x-A-x(3)-[YF]-x(2)-F-[GSA]); purple-dashed box represents chitinases family 19 signature 2 [LIVM]-[GSA]-F-x-[STAG](2)-[LIVMFY]-W-[FY]-W-[LIVM].

**Figure 4 cimb-47-00633-f004:**
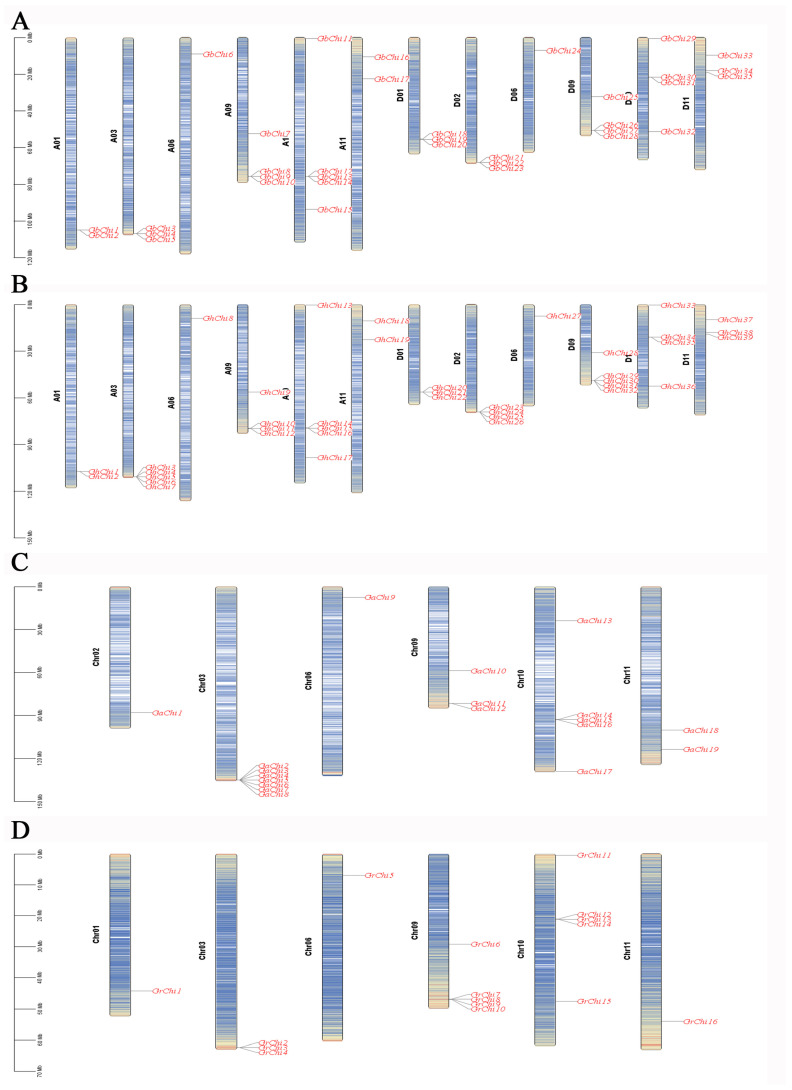
Chromosome mapping of Chi family members in four cotton species. Notes: (**A**) Locations of Chi genes on chromosomes in island cotton; (**B**) locations of Chi genes on chromosomes in upland cotton; (**C**) locations of Chi genes on chromosomes in Asian cotton; (**D**) locations of Chi genes on chromosomes in Raymond cotton. GhA01–GhD11 represent chromosomes 1 to 11 in island cotton, and Chr01–Chr11 represent chromosomes 1 to 11 in Asian and Raymond cotton.

**Figure 5 cimb-47-00633-f005:**
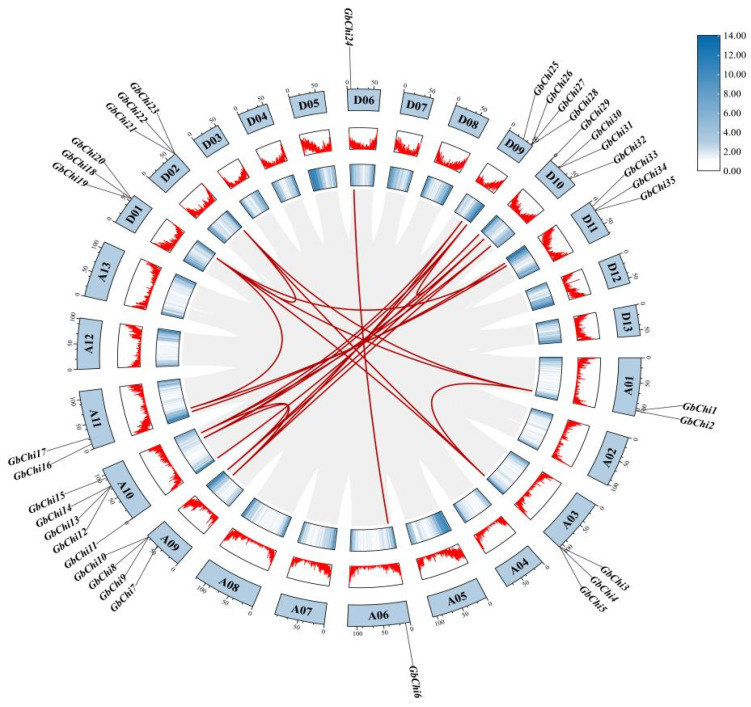
Intra-genomic collinearity of Chi family members of island cotton.

**Figure 6 cimb-47-00633-f006:**
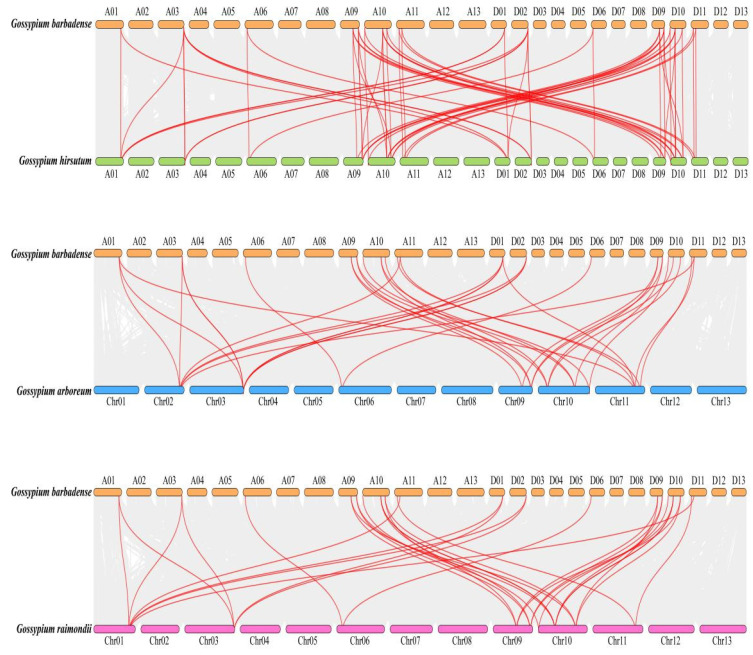
Collinearity of Chi family members among island cotton, upland Asian cotton, and Raymond’s cotton.

**Figure 7 cimb-47-00633-f007:**
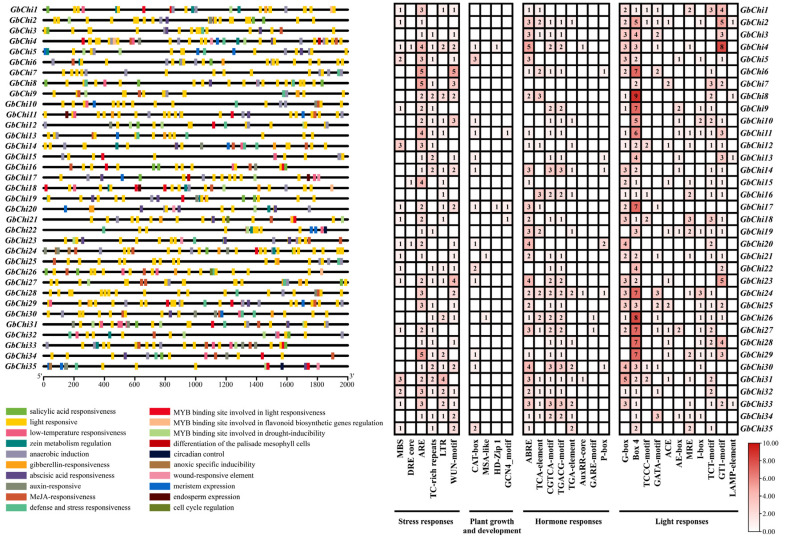
Analysis of promoter cis-elements and motifs of Chi family genes in island cotton.

**Figure 8 cimb-47-00633-f008:**
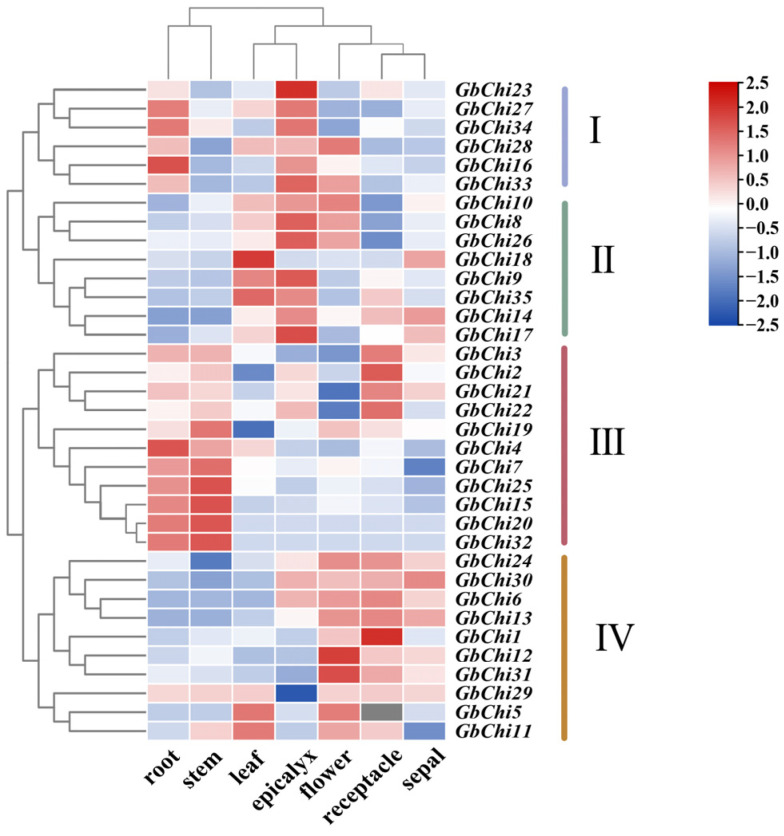
Transcriptional profile analysis and expression levels of GbChi genes in different cotton tissues.

**Figure 9 cimb-47-00633-f009:**
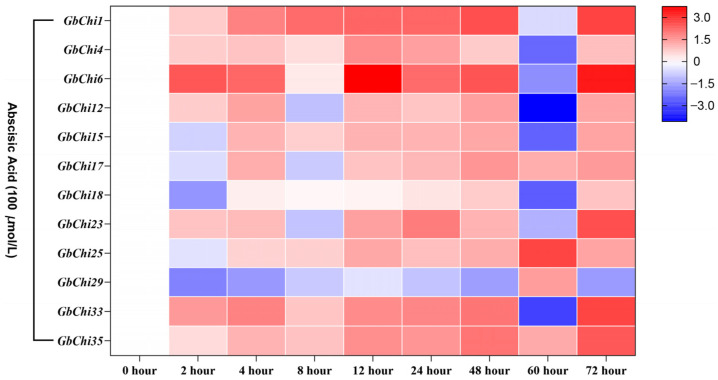
Analysis of the expression profile of the GbChi gene following ABA hormone treatment.

**Figure 10 cimb-47-00633-f010:**
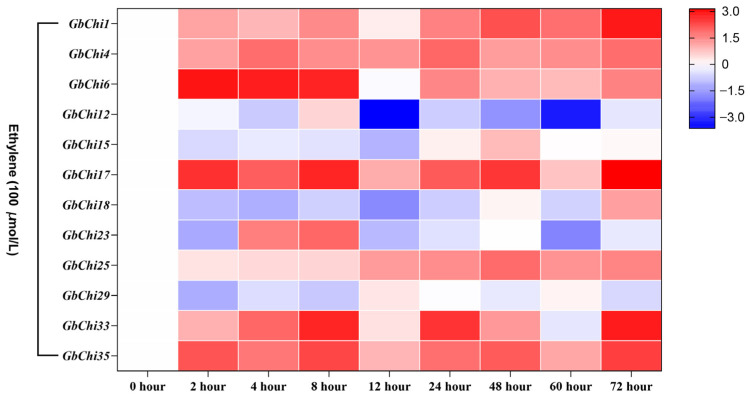
Analysis of the expression profile of GbChi gene under ethylene hormone treatment.

**Figure 11 cimb-47-00633-f011:**
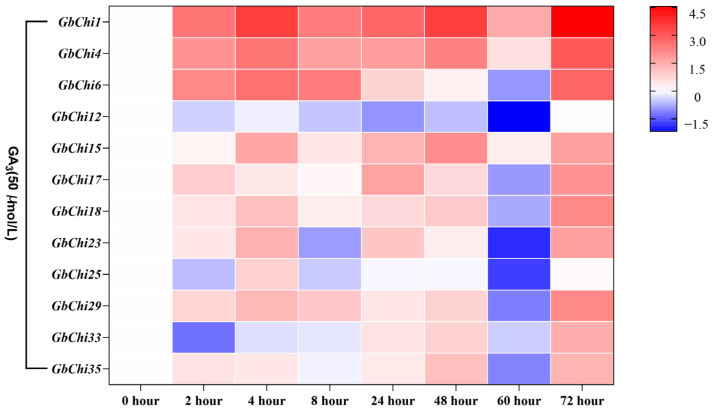
Analysis of the expression profile of GbChi gene under treatment with GA hormone.

**Figure 12 cimb-47-00633-f012:**
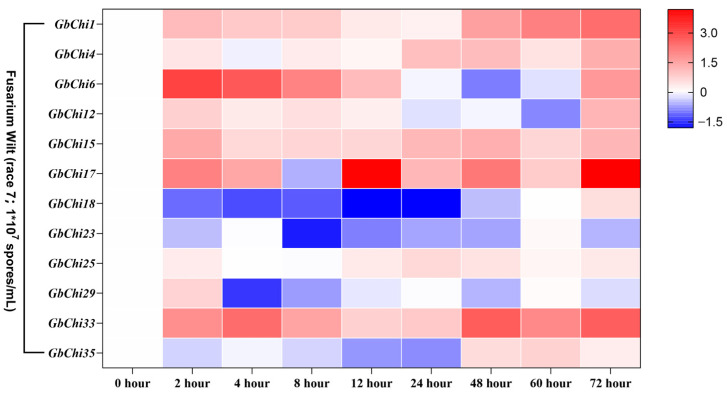
Expression profile of the GbChi gene under fusarium wilt infection.

## Data Availability

Data are contained within the article and Appendix A.

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
